# Conversion of Natural Biowaste into Energy Storage Materials and Estimation of Discharge Capacity through Transfer Learning in Li-Ion Batteries

**DOI:** 10.3390/nano13222963

**Published:** 2023-11-16

**Authors:** Murugan Nanthagopal, Devanadane Mouraliraman, Yu-Ri Han, Chang Won Ho, Josue Obregon, Jae-Yoon Jung, Chang Woo Lee

**Affiliations:** 1Department of Chemical Engineering (Integrated Engineering Program), College of Engineering, Kyung Hee University, 1732 Deogyeong-daero, Giheung, Yongin 17104, Republic of Korea; nanthamurugan1994@gmail.com (M.N.); raman96@khu.ac.kr (D.M.); ghckddnjs@naver.com (C.W.H.); 2Department of Industrial and Management Systems Engineering, Kyung Hee University, 1732 Deogyeong-daero, Giheung, Yongin 17104, Republic of Korea; youry6936@khu.ac.kr; 3Center for the SMART Energy Platform, College of Engineering, Kyung Hee University, 1732 Deogyeong-daero, Giheung, Yongin 17104, Republic of Korea; jobregon@khu.ac.kr; 4Department of Industrial Engineering, Seoul National University of Science and Technology, 232 Gongneung-ro, Nowon-gu, Seoul 01811, Republic of Korea

**Keywords:** lithium-ion battery, biowaste, eggshell membrane, discharge capacity estimation, transfer learning

## Abstract

To simultaneously reduce the cost of environmental treatment of discarded food waste and the cost of energy storage materials, research on biowaste conversion into energy materials is ongoing. This work employs a solid-state thermally assisted synthesis method, transforming natural eggshell membranes (NEM) into nitrogen-doped carbon. The resulting NEM-coated LFP (NEM@LFP) exhibits enhanced electrical and ionic conductivity that can promote the mobility of electrons and Li-ions on the surface of LFP. To identify the optimal synthesis temperature, the synthesis temperature is set to 600, 700, and 800 °C. The NEM@LFP synthesized at 700 °C (NEM 700@LFP) contains the most pyrrolic nitrogen and has the highest ionic and electrical conductivity. When compared to bare LFP, the specific discharge capacity of the material is increased by approximately 16.6% at a current rate of 0.1 C for 50 cycles. In addition, we introduce innovative data-driven experiments to observe trends and estimate the discharge capacity under various temperatures and cycles. These data-driven results corroborate and support our experimental analysis, highlighting the accuracy of our approach. Our work not only contributes to reducing environmental waste but also advances the development of efficient and eco-friendly energy storage materials.

## 1. Introduction

In recent times, powertrain electrification has emerged as a promising option to eliminate greenhouse gases emitted by transport sectors and other types of environmental pollution [1]. Through the Paris Agreement, many nations renewed their policies to promote rechargeable battery-operated electric vehicles (EVs) by minimizing the usage of vehicles with an internal combustion engine [2]. With most countries on the brink of EV validation, Li-ion batteries (LIBs) play a promising role because of their outstanding electrochemical performance [3,4]. The search for batteries with good electrochemical stability and high energy density has laid a platform for more research on LIBs [5,6,7,8]. The most promising cathode materials for LIBs are LiNiCoMnO2 (NCM), LiNiCoAlO_2_ (NCA), LiCoO_2_ (LCO), LiMn_2_O_4_ (LMO), and LiFePO_4_ (LFP) [5,9,10]. Of these, layered cathode materials such as NCM and NCA [11,12,13] seem to deliver a higher specific capacity and wider electrochemical window owing to their increased nickel content and the presence of cobalt [14,15]. Despite their advantages, they are limited owing to the relatively high cost and safety concerns (such as the risk of degradation and thermal instability) associated with them. LCO has a high theoretical specific capacity, high theoretical volumetric capacity, high discharge voltage, and excellent rate capability, which have attracted considerable attention [16,17,18]. However, shortcomings such as the high cost of cobalt, structural changes during intercalation and deintercalation, low cyclic ability, thermal runaway observed at temperatures above 200 °C, which may result from the release of oxygen, and the exothermic reaction between organic materials limit the use of LCO for EVs. To overcome these major limitations, olivine-structured Li iron phosphate (LFP) has been examined for use as a cathode material for LIBs because of its low cost, good thermal stability, and intrinsic safety [19], which seem to be the essential characteristic parameters to be considered when manufacturing batteries for EVs [20,21,22]. Compared to NCM and NCA, LFP delivers a theoretical capacity of 170 mAh g^−1^, a practical capacity in the range 120–160 mAh g^−1^, and an operating voltage range of 3.4–4.5 V, which is relatively moderate; however, LFP provides durability and a long battery life [23,24,25,26].

The natural eggshell membrane (NEM) offers many advantages when applied as an energy storage material [19]. First, NEM is inexpensive and easy to obtain because it is used abundantly in industrial and household fields [27,28]. Second, it is non-toxic and environmentally friendly. Third, NEM contains many natural proteins, such as collagen, which can serve as a nitrogen and carbon source through carbonization and dissolution [29,30,31,32,33].

In LFP, PO_4_^3—^phosphate (a polyanion) captures the tetrahedral sites, whereas Fe^2+^ and Li^+^ occupy lattice positions in octahedral sites where the redox potential increases, leading to the stabilization of the material’s structure. However, LFP possesses low electrical and ionic conductivity and a low average potential compared to layered cathode materials [34]. These drawbacks can be overcome by maintaining the uniform size of the particles (nanosized), doping with foreign cations, and coating the surface of LFP particles with carbon and conductive agents [29,30]. Ultimately, the reduction in particle size results in shortened ionic and electronic transport distances, which allows for the enhancement of rate capability and overall electrochemical performance. Because of the techniques used and the electron carrier efficiency of the carbon source material, the structural and surface conductivities of the material are improved.

In the meantime, recently, data-driven analysis using machine learning has been frequently used to predict system performance in wide fields. Machine learning is a technology that automatically learns data and then predicts the outcomes for given conditions. Particularly, deep learning refers to the use of neural networks (NN) with many hidden layers that are able to uncover non-linear relationships in the data. In the field of battery research, deep learning has been utilized to effectively estimate the charge capacity of LIBs [31]. In general, to accurately predict the charge capacity of a battery, it is required to carry out several experiments with more than 50 charging and discharging cycles. The time to test one charging and discharging cycle varies but is generally long; thus, sometimes it takes several months to collect the desired data [32]. One way to overcome this situation is to use the data from similar batteries in similar experiment settings to pre-train the NN model. It helps to reduce the amount of training data and, hence, the running time of the experiments. This process is known as transfer learning, which is useful when comparing the charging behavior of similar batteries that have various coating materials, and some studies have also been reported for the remaining useful life (RUL) prediction of LIBs [33,34]. We aimed to explore the potential of transfer learning as a predictive tool, demonstrating its applicability and accuracy in estimating battery performance. This focus allowed us to delve deeply into the transfer learning methodology and its implications for battery technology, paving the way for future applications and advancements in the field.

As preliminary research, we conducted a study in which nitrogen-doped carbon derived from biowaste improved both the conductivity on the LFP surface and the electrochemical performance. In this study, we conducted qualitative and quantitative analyses of the nitrogen-doped carbon coating layer formed on the surface of the LFP according to the synthetic temperature and investigated the electrochemical performance of each sample. Furthermore, the discharge capacity of the coated LFP according to the synthesis temperature was predicted using machine learning with cycle data.

## 2. Experimental

### 2.1. NEM Preparation

NEM preparation was carried out as previously reported [19]. In order to remove any remaining albumen, yolk, and other impurities, the natural eggshells were thoroughly washed with distilled water. The washed eggshells were subjected to 18 h of immersion in acetic acid at 4 °C to remove the residual contaminants. Then, NEM was stripped off from the eggshells by neutralizing the immersed eggshells in distilled water. Finally, NEM was contained in a potassium phosphate buffer solution (5 mM, pH 7.0) at 4 °C until before use [19].

### 2.2. Synthesis of NEM@LFP

The preparation of NEM@LFP, a natural eggshell membrane-coated LFP, was synthesized using a calcination-based solid-state approach. NEM powder was obtained by calcining tough NEM fibers at 350 °C for 2 h in an N_2_ atmosphere. Obtaining the powder as carbonized NEM (C-NEM) involved grinding the samples after calcination. LFP with spherical morphology, a particle size of around 2–5 µm, and a density of 3.68 g/cm^3^ was purchased from Sigma Aldrich (St. Louis, MO, USA). Afterward, the commercial LFP and C-NEM (LFP/C-NEM ratio of 100:4) were milled for 6 h to ensure an appropriate particle size and thorough mixing. The final step was calcining the NEM/LFP precursor mixtures at various temperatures (600, 700, and 800 °C) for 3 h in an N_2_ atmosphere; these were named NEM 600@LFP, NEM 700@LFP, and NEM 800@LFP, respectively.

### 2.3. Material Characterization

The effect of coating on the crystal structure was analyzed using X-ray diffraction (XRD, D8 Advance, Bruker, Billerica, MA, USA) with CuKα radiation at a wavelength of 1.5405 Å. X-ray photoelectron spectroscopy (XPS, K-alpha, Thermo Fisher, Waltham, MA, USA) was employed to perform a chemical state analysis and quantitative analysis to determine its composition.

### 2.4. Electrochemical Measurements

The electrochemical performances of both the bare LFP and NEM@LFP materials were analyzed through a CR2032-type coin cell. The working electrode was fabricated by the homogenous mixing of LFP and NEM@LFP as the active material, Denka Black as a conductive agent, and polyvinylidene difluoride (PVDF) as a binder in the weight ratio 85:10:5 while employing N-methyl pyrrolidone (NMP) as a disperser to obtain a slurry of appropriate viscosity. The slurry obtained was then coated on an aluminium foil with a mass loading of 1.8 mg cm^−2^. The slurry coated on aluminium foil was allowed to dry naturally overnight, and then it was dried for 5 h at 120 °C in a hot air oven. The dried electrode underwent a process of roll-pressing and was cut into a 14 mm diameter size, followed by vacuum oven drying at 120 °C for 5 h. Using an argon gas-filled glove box, the assembly of the coin cell was executed, consisting of pure Li foil (as the reference electrode), an as-prepared LFP or NEM@LFP electrode as a working electrode, Celgard 2340 as a separator, and 1M LiPF_6_ dissolved in an EC/DEC (1:1 volume ratio) as an electrolyte. Assembling the electrodes and separator with a few drops of electrolyte, they were then crimped and tested using a cycler (Battery Tester 05001, HTC, Hwaseong, Republic of Korea). The electrochemical workstation (Iviumstat, Ivium Technologies) was used to perform electrochemical impedance spectroscopy (EIS) over the frequency range of 100 kHz to 10 mHz. At room temperature, all the electrochemical performance was performed, and all measurements were conducted at room temperature to ensure reliability.

### 2.5. Discharge Capacity Estimation Using Transfer Learning

In transfer learning, the weights of the NN trained for one task are used to warm up the training of another network. This allows the second network to be trained with less data and obtain good results. Internally, some specific layers among multiple layers in the network are “frozen” through pre-training their weights, and, therefore, those layers are not retrained. After that, the remaining layers are retrained with additional data. Using this method, it is possible to reflect both the features of the already learned data and the features of the newly learned data [35,36,37].

The data-driven experimental setting for discharge capacity estimation using transfer learning is presented in Figure 1. It uses the data of 93 charging-discharging cycles of bare LFP materials for pre-training a NN. The input to the NN model is the voltage and current for each cycle, and the output of the network is the resulting discharge capacity. The NN includes four hidden layers with 500 neurons and with the ‘relu’ activation function. Using transfer learning, we retrained only the last layer of three predictive models for the NEM materials, one for each synthesis temperature: 600@LFP, 700@LFP, and 800@LFP. The date was divided for training and testing to check the prediction performance of the trained NN; the data of the first 10 cycles was used for training in the transfer learning stage, and the data of the remaining 90 cycles was used for testing. The performance for discharge capacity estimation was evaluated using the mean absolute error (MAE) and the mean absolute percentage error (MAPE).

## 3. Results and Discussion

The crystal structures of bare LFP and as-prepared ESM@LFP materials were investigated through X-ray diffraction (XRD) analysis, and the resulting diffraction patterns are shown in Figure 2. All the patterns could be identified as the orthorhombic olivine phase LFP (JCPDS no. 40–1499), as reported in previous studies [38,39]. However, in the XRD pattern of bare LFP, unidentified peaks were observed. These peaks are presumed to be caused by unspecified additives present in the LFP obtained from an external chemical supplier. Interestingly, during the thermal decomposition process of the nitrogen-doped carbon coating in ESM@LFP, these unknown peaks disappeared completely from the XRD pattern. Notably, the diffraction patterns did not reveal the presence of the nitrogen-doped carbon phase, likely due to its low content. This observation indicates that the ESM coating does not influence the crystal structure, suggesting that the structural integrity of LFP is maintained even after the application of the ESM coating.

The quantitative analysis and chemical state of nitrogen-doped, carbon-coated LFPs were elucidated through XPS analysis. The presence of a nitrogen peak in the NEM@LFP survey spectra presented in Figure 3a is evidence that nitrogen-doped carbon was produced from NEM. As projected in Figure 3b–d, the presence of multiplex profiles for the N1s orbital of NEM@LFP was determined to be pyridinic nitrogen (N-6) (397.6–398.4 eV), pyrrolic nitrogen (N-5) (399.7–400.8 eV), and oxidized pyridinic nitrogen (N-X) (401–405 eV) [40,41,42]. The contents of pyridinic, pyrrolic, and oxidized pyridinic nitrogen in NEM 600@LFP were found to be 52.41%, 42.51%, and 5.08%, respectively, whereas those of NEM 700@LFP were 59.13%, 37.39%, and 3.48%, and NEM 800@LFP were 43.98%, 50.07%, and 5.95%, respectively, as displayed in Table 1. The formation and distribution of nitrogen species in carbon materials, especially pyrrolic nitrogen, are highly influenced by the synthesis temperature. Higher temperatures can facilitate the incorporation of nitrogen into the carbon matrix in various forms, leading to differences in the pyrrolic nitrogen content observed in the study. In addition, the existence of different nitrogen contents promotes ionic and electrical conductivity, providing more active sites for Li movement. The existence of these nitrogen profiles arises from the breakdown of collagen, which is the major part of NEM. Of the three nitrogen profiles, pyrrolic nitrogen seems to be most effective for electrical and ionic conductivity owing to the presence of more π-bonding, which promotes the movement of Li-ions and electrons. Table 1 shows that NEM 700@LFP has the highest percentage of pyrrolic nitrogen, from which it can also be expected to have the most efficient electrochemical performance.

The formation and distribution of nitrogen species in carbon materials, especially pyrrolic nitrogen, are highly influenced by the synthesis temperature. Higher temperatures can facilitate the incorporation of nitrogen into the carbon matrix in various forms, leading to differences in the pyrrolic nitrogen content observed in the study. In addition, the existence of different nitrogen contents promotes ionic and electrical conductivity, providing more active sites for Li movement. The existence of these nitrogen profiles arises from the breakdown of collagen, which is a major part of NEM. Of the three nitrogen profiles, pyrrolic nitrogen seems to be most effective for electrical and ionic conductivity owing to the presence of more π-bonding, which promotes the movement of Li-ions and electrons. Table 1 shows that NEM 700@LFP has the highest percentage of pyrrolic nitrogen, from which it can also be expected to have the most efficient electrochemical performance.

The Nyquist plot displayed in Figure 4, derived from electrochemical impedance spectroscopy (EIS) data, offers critical insights into the electrochemical behavior of bare LFP and NEM@LFP. The Rs component, signifying electrolyte ohmic resistance, the semicircle denoting charge transfer resistance (R_ct_), and the presence of slanting lines representing the Warburg resistance of Li-ion diffusion collectively elucidate the intricate interfacial processes within the electrodes. In Table 2, the remarkably low R_s_ and R_ct_ values for NEM 700@LFP, aligning with its highest pyrrolic nitrogen content (as indicated in Table 1), underscore the pivotal role of nitrogen dopants. These dopants not only optimize the charge transfer interfaces but also contribute to the creation of defects within the carbon structure, enhancing disorder. 

This disorder promotes efficient charge transfer kinetics, facilitating the rapid movement of electrons and Li-ions. The synergistic effects of increased disorder and nitrogen incorporation fostered by the nitrogen-doped carbon coating in NEM 700@LFP highlight a dynamic interplay of factors that ultimately augment the efficiency of charge-transfer reactions. These findings not only underscore the significance of nitrogen content in dictating electrochemical performance but also illuminate the nuanced impact of carbon disorder, providing valuable insights into the design principles of advanced electrode materials for high-performance energy storage systems [43].

Between a potential range of 2.5 and 4.2 V, the electrochemical behaviors of bare LFP and NEM 700@LFP were examined. Figure 5a shows the differential capacity vs. voltage curve for the first cycle of the two electrodes. The NEM 700@LFP exhibited sharp redox peaks compared to the bare electrode within a voltage interval of approximately 0.076 V, whereas for bare LFP it was approximately 0.085 V. These results confirm that the reversibility and kinetics of Li intercalation/deintercalation can be improved by coating with NEM. Figure 5b–c illustrate the galvanostatic profiles of the charge–discharge curves measured at a 0.1 C current rate for bare LFP and NEM 700@LFP, respectively. Both materials demonstrate a plateau at around ~3.4 V, which corresponds to the two-phase redox reaction involving LiFePO_4_ and FePO_4_. The NEM 700@LFP delivers a discharge capacity of approximately 156.9 mAh g^−1^, which is higher than the discharge capacity of 132.4 mAh g^−1^ for bare LFP during the initial cycle. The enhanced discharge capacity of NEM 700@LFP reveals the effective coating of nitrogen-doped carbon on the LFP. However, after 50 cycles, the discharge capacities of NEM 700@LFP and bare LFP were obtained as 157.2 and 134.8 mAh g^−1^, respectively. Incorporating nitrogen into the carbon coatings enhances both electronic and ionic conductivity by introducing additional charge carriers and creating pathways for electrons and Li-ions, ensuring efficient charge and discharge processes. Furthermore, these nitrogen dopants optimize the surface chemistry of LFP particles, providing active sites for Li-ion adsorption and desorption, thus improving electrochemical kinetics. Additionally, the nitrogen dopants contribute to the formation of a stable solid-electrolyte interface (SEI) layer, minimizing electrolyte decomposition and enhancing long-term cycling stability. Moreover, the facilitated Li-ion diffusion, facilitated by these nitrogen dopants, enables rapid movement of Li-ion between crystal lattice sites, leading to high specific discharge capacities. Thus, compared to bare LFP, NEM 700@LFP was found to offer better electrochemical performance due to its nitrogen-doped carbon coating.

The results of the data-driven analysis are presented in Table 3 and Figure 6. The discharge capacity estimation of NEM @LFP based on the trained NN models is presented in Figure 6. Using transfer learning, the discharge capacity at a 0.1 C rate was predicted for each synthesis temperature. Three NN models (NEM 600@LFP, NEM 700@LFP, and NEM 800@LFP) were trained, and those models showed that NEM 700@LFP would have the highest capacity, which agrees with the findings of the chemical analysis. MAPE was kept between 1% and 4% for all the experiments. The specific discharge capacities obtained from our experimental analysis of the NEM 700@LFP material closely match the capacity values estimated by the neural network NN models. This congruence between the experimental data and the NN model predictions underscores the efficacy of transfer learning in accurately estimating battery performance. The successful alignment of real-world data with model predictions signifies a significant milestone in the field of battery technology. This robust agreement not only confirms the reliability of our transfer learning approach but also holds promising implications for practical applications. Accurate predictions derived from transfer learning empower us to make informed decisions regarding the deployment of batteries in various settings. By precisely estimating the battery’s performance, we can optimize its utilization in specific applications where its characteristics align perfectly. This precision ensures efficient resource utilization, cost-effectiveness, and sustainable energy practices.

## 4. Conclusions

The successful conversion of natural eggshell membrane, a biowaste, into a coating additive material for energy storage was achieved. Through solid-state synthesis, biowaste-derived nitrogen-doped carbon was coated on the surface of NEM 600@LFP, NEM 700@LFP, and NEM 800@LFP, consisting of pyridinic nitrogen, pyrrolic nitrogen, and oxidized pyridinic nitrogen. NEM 700@LFP had the highest pyrrolic nitrogen content, which meant that it also had the highest ionic and electrical conductivity. The EIS analysis also showed the smallest R_ct_ for NEM 700@LFP. In addition, when the capacity of the synthetic temperature was predicted through data-driven analysis, the capacity at 700 °C was predicted to be the highest. NEM 700@LFP also showed improved capacity and electrochemical performance compared to its counterparts. The nitrogen-doped carbon coating layer promoted the transportation of electrons and Li-ions between the electrolyte and electrode.

## Figures and Tables

**Figure 1 nanomaterials-13-02963-f001:**
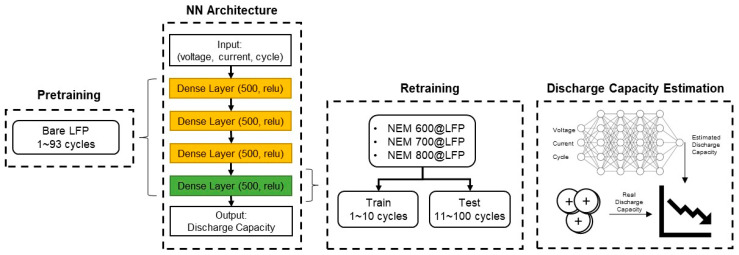
Data-driven experiment setting for estimating the discharge capacity of NEM@LFP cells using transfer learning from models trained with data from bare LFP cells.

**Figure 2 nanomaterials-13-02963-f002:**
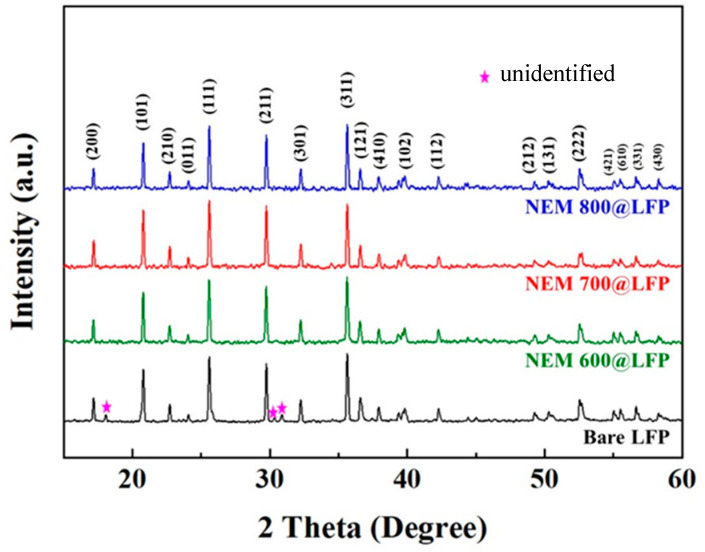
XRD patterns of bare LFP, NEM 600@LFP, NEM 700@LFP, and NEM 800@LFP, respectively.

**Figure 3 nanomaterials-13-02963-f003:**
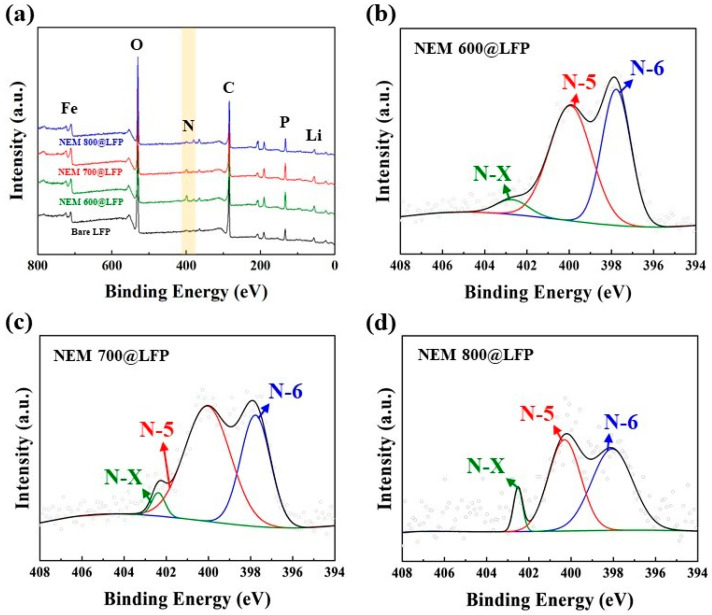
(**a**) XPS survey spectra of bare LFP, NEM 600@LFP, NEM 700@LFP, and NEM 800@LFP and XPS multiplex survey for N 1s orbital of (**b**) NEM 600@LFP, (**c**) NEM 700@LFP, and (**d**) NEM 800@LFP, respectively.

**Figure 4 nanomaterials-13-02963-f004:**
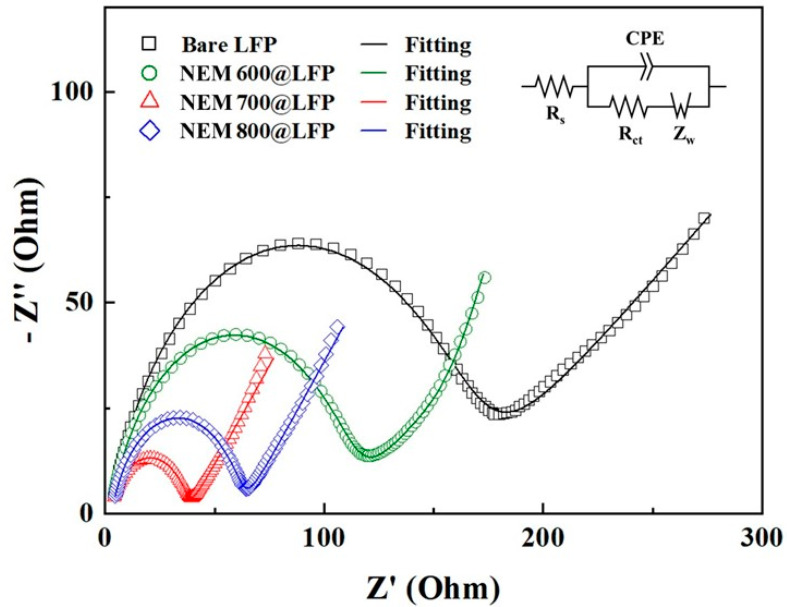
Nyquist plots of bare LFP, NEM 600@LFP, NEM 700@LFP, and NEM 800@LFP electrodes in fresh cells with fitting curves by using the inset equivalent circuit.

**Figure 5 nanomaterials-13-02963-f005:**
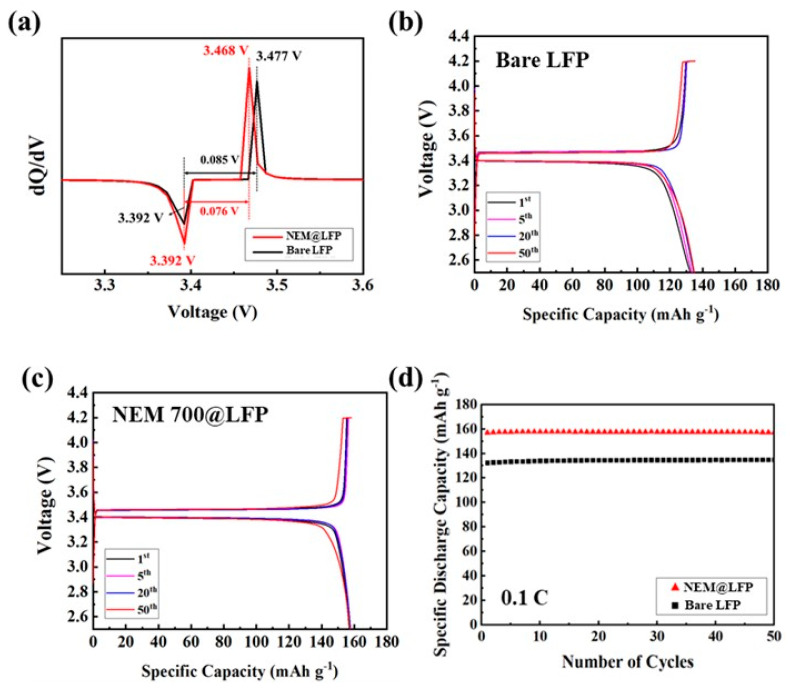
(**a**) The differential capacity vs. voltage curve for the first cycle of bare LFP and NEM 700@LFP, the galvanostatic potential profiles for (**b**) bare LFP, (**c**) NEM 700@LFP, and (**d**) cycling performance profiles showing discharge capacity of bare LFP and NEM 700@LFP for 50 cycles between the voltage window of 2.5 to 4.2 V at a current rate of 0.1 C.

**Figure 6 nanomaterials-13-02963-f006:**
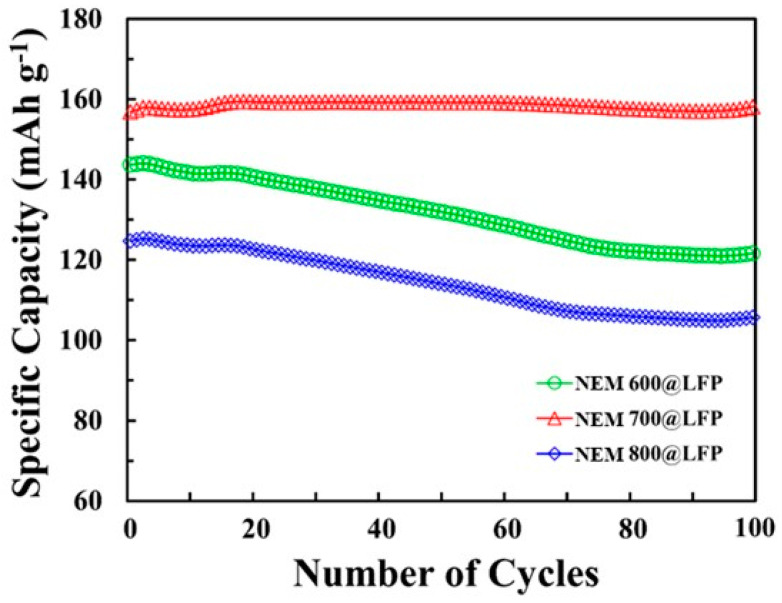
Estimated discharge capacity according to the number of cycles using the retrained NN models, NEM 600@LFP, NEM 700@LFP, and NEM 800@LFP.

**Table 1 nanomaterials-13-02963-t001:** The percentage content of different chemical states of nitrogen in XPS.

Sample Code	N-6 (%)	N-5 (%)	N-X (%)
NEM 600@LFP	42.51	52.41	5.08
NEM 700@LFP	37.39	59.13	3.48
NEM 800@LFP	50.07	43.98	5.95

**Table 2 nanomaterials-13-02963-t002:** The fitted parameters of bare LFP, NEM 600@LFP, NEM 700@LFP, and NEM 800@LFP, respectively.

Sample Code	Fitted Parameters	
	R_s_ (Ω)	R_ct_ (Ω)
Bare LFP	2.9	157.5
NEM 600@LFP	2.6	107
NEM 700@LFP	2.5	35.4
NEM 800@LFP	3.1	59.1

**Table 3 nanomaterials-13-02963-t003:** Prediction performance of trained NN models for discharge capacity estimation.

NN Model	MAE	MAPE
Bare LFP	1.7895	0.0131
NEM 700@LFP	6.4702	0.0431
NEM 800@LFP	2.5696	0.0226

## Data Availability

Data are contained within the article.

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
