# Peer review of "Conversion of Natural Biowaste into Energy Storage Materials and Estimation of Discharge Capacity through Transfer Learning in Li-Ion Batteries"

_nanomaterials, 2023, doi:10.3390/nano13222963_

Round 1
Reviewer 1 Report
Comments and Suggestions for Authors
In this submission, the authors report the synthesis of composite NEM@LFP using the natural eggshell membrane and LFP. This material shows high performance for LIB. Data-driven analysis was also performed to show its enhanced performance. The topic is interesting and could attract a wide readership from researchers working in the nanomaterials area. Therefore, I recommend its publication after the following issues are addressed.
1. How do the authors obtain LFT?
2. There is no XRD data, which is essential to reveal the material’s crystal structure and composition.
3. There is no SEM and/or TEM images for the electrode materials. The microstructure could greatly affect the battery performance.
4. The capacity estimated via the transfer learning approach could be compared with the experimental results.
5. The results and discussion part is too weak. Some figures has no detailed discussions.
6. The authors are recommended to cite relevant literatures such as ACS Appl. Mater. Interfaces 2020, 12, 15120−15127; Ceram. Int. 2019, 45, 18614–18622; ACS Sustainable Chem. Eng. 2020, 8, 17106−17115.

Comments on the Quality of English Language
N. A.
Author Response
We thank you for your thorough review and highly welcome the comments and suggestions, which significantly contributed to improving the quality of the manuscript. The following responses (the response parts are highlighted in blue colour) have been prepared to address the reviewer’s comments.

Reviewer 2 Report
Comments and Suggestions for Authors
The authors reported a nitrogen-doped carbon derived from natural eggshell membranes (NEM) to coat LiFePO4 through the solid-state thermal treatment, which can improve the electrical and ionic conductivity of LiFePO4 cathode of Lithium ion battery. The effect on the qualitative and quantitative properties of the nitrogen-doped carbon coating layer derived from NEM at 600 ℃, 700 ℃ and 800 ℃ is investigated using XPS and EIS. And the NEM@LFP obtained at 700℃ exhibited larger specific capacity than pare LiFePO4. Additionally, the capacity and cycling performance was predicted using machine learning. I recommend that this manuscript can be accepted for publication in nanomaterials after the following revision has been carefully done.
1. To demonstrate the LiFePO4 was successfully coated by nitrogen-doped carbon, the corresponding SEM and TEM of NEM@LFP should be provided.
2. In Figure 4a, why is the CV curves so steep instead of smooth? This may be because that record points of data are too few. The CV curves need to be re-collected.
3. The XPS results show that the pyrrolic nitrogen content is different at different temperatures. Please explain the reasons for this. Furthermore, the electrochemical property of alkali metals should be discussed in the Introduction part and the review (Advanced Materials, 2022, 34, 2108432) should be referred and cited.
4. In this work, the data of electrochemical performance is incomplete. The rate capability and long-life cycle performance should be given.
5. There are some mistakes in this manuscript, like “handIn the meantime, recently data-driven …”. Please carefully check throughout the manuscript.
Author Response

(The authors gave the same response as above.)

Reviewer 3 Report
Comments and Suggestions for Authors
The authors should address the following comments in detail for further processing,
1. Similar approach (in material preparation) for the similar Battery system Li-LFP system studied and published. ( same group). So, please mention and highlight its significance.( if followed similar approach should address the same)
Thermally assisted conversion of biowaste into environment-friendly energy storage materials for lithium-ion batteries
https://doi.org/10.1016/j.chemosphere.2021.131654
2. The abstract should be revised. (too descriptive make it concise and comprehensive)
3. In Fig 4.d and Fig.5 data for NEM 700@LFP is not the same why? and explain.
4. For Fig.5 include standard sample data for comparison.
5. Explain why the performance shift is so high compared to the bare Explain. In other words, No specific trend is explained (for Fig.5).
6. Possible mechanisms or discussions should be elaborated.
7. Writing should be improved, revise results and discussion.
Comments on the Quality of English Language
Writing is not impressive and lengthy.
Author Response

(The authors gave the same response as above.)

Round 2
Reviewer 2 Report
Comments and Suggestions for Authors
The authors have well addressed my concerns and I would like to recommend it to be accepted in the esteemed journal "Nanomaterials".
Reviewer 3 Report
Comments and Suggestions for Authors
The authors responded to all comments with appropriate discussion and details and also included them in the revised manuscript. The revised manuscript was significantly improved by the authors.